# Higher Embedding Dimension Creates a Stronger World Model for a Simple Sorting Task

## Abstract

We investigate how embedding dimension affects the emergence of an internal "world model" in a transformer trained with reinforcement learning to perform bubble-sort-style adjacent swaps. Models achieve high accuracy even with very small embedding dimensions, but larger dimensions yield more faithful, consistent, and robust internal representations. In particular, higher embedding dimensions strengthen the formation of structured internal representation and lead to better interpretability. After hundreds of experiments, we observe two consistent mechanisms: (1) the last row of the attention weight matrix monotonically encodes the global ordering of tokens; and (2) the selected transposition aligns with the largest adjacent difference of these encoded values. Our results provide quantitative evidence that transformers build structured internal world models and that model size improves representation quality in addition to end performance. We release our metrics and analyses, which can be used to probe similar algorithmic tasks.

## 1 Introduction

Sorting has long been a canonical problem in computer science, valued both for its theoretical significance and its ubiquity in practice Cormen et al. (2022). Beyond efficiency concerns, sorting also serves as a natural testbed for studying how algorithms—and, increasingly, learned models—structure computation. Recent advances have shown that deep learning can yield new insights into traditional computer science problems such as sorting, even leading to improvements that have been incorporated into LLVM standard sort for C++ Mankowitz et al. (2023). Our interest, however, does not lie in algorithmic acceleration, but rather the mechanisms by which small neural architectures internally represent and execute discrete procedures.

Transformers are now the dominant architecture for sequence modeling Vaswani et al. (2017), achieving remarkable capabilities in language, reasoning, and beyond Li et al. (2023). Logical and mathematical benchmarks highlight their strengths but also reveal persistent reasoning limitations Cobbe et al. (2021). While strategies such as chain-of-thought prompting mitigate some failures Zhang et al. (2023a); Wei et al. (2022), these remain workarounds rather than evidence of robust reasoning. In parallel, mechanistic interpretability has emerged as a program for understanding the inner workings of neural networks Chughtai et al. (2023). Small transformer networks without multilayer perceptrons in particular have been found to be fairly amenable to analysis Elhage et al. (2021). Related work on grokking shows that transformers can form generalizable latent representations of tasks Liu et al. (2022). This behavior connects to the broader "world model" hypothesis: neural networks build structured internal representations of their environment Ha & Schmidhuber (2018). Encouraging the development of a world model has been shown to improve reasoning abilities in large language models (LLMs) Hao et al. (2023). We consider a minimalist version of this hypothesis, where the learned representation captures the essential properties of the environment's state.

In this paper, we investigate whether small transformers, trained via reinforcement learning (RL), develop an interpretable world model of the bubble sort process and how this model's fidelity varies with embedding dimension. Instead of phrasing it as an autoregressive generation problem, we

recast it in a RL framework and take advantage of the interpretability of the transformer to gain insight into how the model solves the task. Our RL setting uses permutations of tokens as states, with adjacent swaps as the only allowed actions. The model is rewarded for producing a sorted sequence and is trained using the Proximal Policy Optimization (PPO) algorithm Schulman et al. (2017). Importantly, the environment only incentivizes agents to eventually sort a sequence, not to discover the *optimal* sorting path. The optimal solution is not unique, and suboptimal but valid sorting strategies may still achieve perfect reward. Our analysis therefore focuses on the internal mechanisms agents converge to, rather than on efficiency.

Our contributions are as follows:

- We introduce sorting as a reinforcement learning testbed for mechanistic interpretability.
- We provide an empirical study showing that increasing embedding dimension substantially improves representation faithfulness, consistency, and robustness.
- We identify two consistent patterns—a global order encoding in attention weights and largest difference selection rule—that explain agent behavior.
- We release metrics and methodology for evaluating alignment between hypothesized mechanisms and model internals.

## 2 RELATED WORKS

### 2.1 TRANSFORMER-BASED WORLD MODELS IN REINFORCEMENT LEARNING

Recent research has explored the use of transformer architectures to build world models that enhance data efficiency in RL. For example, IRIS combines a discrete autoencoder with an autoregressive transformer to learn efficient world models capable of achieving human-level performance on Atari within only two hours of gameplay Micheli et al. (2022). Similarly, STORM integrates stochastic transformers with variational components to enhance model-based RL Zhang et al. (2023b).

### 2.2 MECHANISTIC INTERPRETABILITY IN NEURAL SYSTEMS AND TOY ENVIRONMENTS

Mechanistic interpretability refers to identifying internal, algorithmically meaningful structures within models. Landmark studies in transformer interpretability, such as those by the Anthropic Clarity team, have revealed algorithmic behaviors, such as induction heads, in small-scale transformers Olsson et al. (2022). More recent work, including analysis of Othello-GPT, demonstrates that linear latent world representations can form even without explicit supervision Nanda et al. (2023). In parallel, studies such as SortBench evaluate how well large language models can capture and execute simple algorithmic behaviors like sorting Herbold (2025).

### 2.3 REPRESENTATION CAPACITY AND EMBEDDING DIMENSION

The question of how model capacity—especially embedding size—affects representation quality has been widely studied. For example, Word2vec models exhibit improved embedding quality with increased dimensionality, up to a point of diminishing returns Mikolov et al. (2013). Empirical studies show that higher-dimensional embeddings often enhance intrinsic task performance, though extrinsic tasks may require careful tuning Melamud et al. (2016). Theoretical work further reveals a bias–variance trade-off that shapes optimal embedding dimensionality Yin & Shen (2018).

## 3 PRELIMINARIES AND DISCUSSIONS

### 3.1 TRANSFORMERS

Transformers Vaswani et al. (2017) have become the dominant architecture for sequence modeling, excelling in language, vision, and RL. Their core mechanism, self-attention, computes interactions between all pairs of tokens in a sequence, enabling flexible representation of order and relational structure. While modern applications often rely on deep, multi-head architectures, even single-head, shallow transformers are capable of learning structured, interpretable behaviors. This makes them

ideal candidates for mechanistic interpretability studies in constrained settings. In this work, we deliberately use minimal transformer architectures to isolate the role of embedding dimension in shaping internal representations (details of the model architecture are provided in Appendix A.1). By reducing architectural complexity, we can more directly attribute observed world-model–like behavior to the interaction of self-attention and RL dynamics.

## 3.2 PROXIMAL POLICY OPTIMIZATION

Proximal Policy Optimization (PPO) Schulman et al. (2017) is a RL algorithm widely adopted for its balance between stability and performance. PPO constrains policy updates using a clipped surrogate objective, which prevents destructive shifts in behavior while still encouraging steady improvement. Its simplicity and robustness have made it a standard choice across domains ranging from Atari to robotics, and more recently as the backbone of RL with human feedback in large language models. In our setting, PPO provides a practical way to train agents to discover sorting strategies without prescribing explicit rules. Because PPO naturally encourages agents to form structured latent representations of their environment, while maintaining training stability, it is particularly well-suited for probing whether interpretable world models emerge in a toy sorting task.

## 3.3 NOTATION AND MODEL SETUP

The input to the model consists of tokens corresponding to numerical values from $1$ to $\ell$, ordered by some permutation $\pi$. The goal is to find a sequence of adjacent swaps that transforms this permutation into a sorted sequence. An adjacent swap operation at index $i$ involves swapping the elements at position $i$ and $i + 1$. The transformer uses single-headed self-attention, with queries $Q$, keys $K$ of dimension $d_k$, and values $V$. With this model, the attention weights are given by

$$W = \frac{QK^\top}{\sqrt{d_k}}.$$

## 3.4 WHY SORTING AS A TESTBED

Our work builds upon a growing body of research that uses toy problems as controlled settings for mechanistic interpretability studies. By constraining the complexity of the task and environment, researchers can systematically probe model internals without the confounding variability present in real-world data Elhage et al. (2021); Chughtai et al. (2023). Sorting, despite its apparent simplicity, is a particularly appealing choice because its state space is finite, its optimal solutions are well-defined, and its intermediate states have a clear, human-interpretable structure. This makes it possible to directly compare the agent's learned representations to ground truth abstractions such as "global order" or "adjacent differences." Unlike real-world RL tasks, there is no ambiguity in goal definition or reward signals, which allows us to attribute observed behaviors more confidently to architectural properties rather than to environmental noise.

Unlike prior work that uses deep RL to search for efficient new algorithms, our aim is not to improve performance but to understand the mechanisms a network employs to solve a task it could already solve Mankowitz et al. (2023). We deliberately choose to use a RL setup because RL-trained models tend to develop internal state representations better aligned with an environment's causal structure, resonating with the "world model" hypothesis in deep learning Ha & Schmidhuber (2018). In our case, the emergent ordering circuit in the attention weights can be seen as a minimalist form of such a model.

Furthermore, there is precedent for embedding dimension playing a critical role in representation quality across modalities. In natural language processing, larger hidden sizes often permit more nuanced syntactic and semantic encoding Hong et al. (2024). By exploring this axis in a highly constrained RL setting, we aim to isolate how embedding dimension affects the fidelity of an interpretable, algorithmically relevant latent structure.

# 4 METHODS

## 4.1 EXPERIMENTAL DESIGN

We trained agents with embedding dimensions varying from 2 to 128 and sequence lengths of 6 and 8. In total, 475 agents were trained, with multiple random seeds, enabling a robust estimate of both mean performance and variability, which we later connect to embedding dimension. Length 6 sequences have a relatively small permutation space (720 possible states), making convergence feasible for nearly all runs, while length 8 (40,320 possible states) introduces enough combinatorial growth without becoming computationally prohibitive.

Embedding dimension was chosen as our primary independent variable because it directly controls the representational capacity of the attention mechanism. This choice also allowed us to probe the trade-off between minimal capacity sufficient for high performance and excess capacity that may encourage structured representations. In principle, embedding dimension directly modulates representational capacity. A larger query/key dimension enhances the expressive power of attention, allowing the network to approximate complex functions Amsel et al. (2024). Additionally, increasing embedding dimensionality enriches the granularity of token embeddings, which empirically improves generalization.

Notably, while we tested a large range of embedding dimensions, the results generally only changed below an embedding dimension of approximately 30 and leveled out after that. There were also some differences in results for the different sequence lengths which should be investigated further, but this was not the main focus of the experiment and would require much more computation because of the combinatorially increasing state space.

## 4.2 IMPLEMENTATION DETAILS

All models were stripped down transformers with an embedding layer, a single-head self-attention block, and a linear layer. They used a decoder-only, GPT-style causal design with learned token and position embeddings. This choice is crucial as it means each token can only attend to previous tokens in the sequence. The model outputs a single hidden embedding which is then fed into separate linear layers for the actor and critic heads in the PPO algorithm.

Agents were trained using the Proximal Policy Optimization (PPO) algorithm, which was chosen for its balance between stability and sample efficiency; it avoids catastrophic policy updates via clipping while remaining easy to parallelize Guo et al. (2025). A learning rate of 2.5e-4 and a discount factor of 0.99 were used (full hyperparameters are in Appendix A.2). The agent receives a reward of +1 if the permutation is sorted after making a transposition and a reward of -0.001 otherwise. Training was performed for 1M, 2M, or 10M timesteps. Training was performed on Nvidia H100 GPUs.

## 4.3 MODEL EVALUATION

To quantify how consistently the above two observed mechanisms appear across embedding dimensions, we define three metrics:

1. **Sorting accuracy:** For each possible initial permutation, we verify that the choice of transposition is a correct swap. Accuracy is defined as the fraction of correct swaps, which allows us to calculate an accuracy between 0 and 1. If an agent has an accuracy $= 1$, it will be able to sort any permutation optimally.

2. **Global order encoding:** Consider the attention weights $W = QK^{\top}/\sqrt{d_k}$, with final row $W_\ell$. We compute the proportion of non-inversions between $(\pi(1), \ldots, \pi(n))$ and $(W_{\ell,1}, \ldots, W_{\ell,n})$, where an inversion is defined as a pair of elements which are out of order between the two sequences. We then normalize by $\binom{n}{2}$, the maximum number of inversions between two sequences. This yields a metric in $[0, 1]$, with 0 meaning the attention weights are completely out of order, and with 1 indicating perfect alignment.

3. **Difference-based swap rule:** For every initial permutation, we sort the differences between consecutive attention output values. We then find the ranking of the transposition which was actually chosen according to this sorted list. We measure the frequency with which the chosen

swap lies among the top-$k$ predicted differences, so a top 2 proportion of 1 would mean all swaps chosen by the agent were in the top 2 (of $n$) predictions.

## 5 RESULTS

We observed that agents consistently converged to a simple and interpretable algorithm when solving the sorting task. Specifically, two mechanisms emerge consistently across runs:

1. **A global order encoding in attention weights:** the final row of the attention weight matrix maps each number token to a value between 0 and 1, and lower attention weight values correspond to lower numerical tokens. This means that the model discovers the global ordering of the input tokens (Observation 1)

2. **A difference-based rule for swap selection:** The value matrix and final linear layer of the network calculate the difference between consecutive attention weight values, and the chosen transposition corresponds to the largest difference (Observation 2)

We hypothesize that together, these two observations define a simple circuit underlying sorting behavior. Importantly, while small embedding dimensions are sufficient for near-perfect accuracy, larger embedding dimensions make these mechanisms more consistent, more faithful, and more robust across agents. In the following subsections, we show evidence that agents reliably learn this circuit, and that convergence to it becomes more consistent as embedding dimension increases.

### 5.1 ACCURACY OF AGENTS

Agents reliably learned to sort sequences under a wide range of embedding dimensions. For sequence length 6, 99.2% of the agents reliably achieved 100% accuracy once the embedding dimension was greater than 16. For the sequence length of 8, many agents still achieved 100% accuracy, but at a lower rate of 37.4% when the embedding dimension was above 16. Our observations relate to models which achieve a perfect or near-perfect accuracy, and the PPO training algorithm was able to achieve this for many agents at each length. The average accuracy at each embedding dimension is shown in Figure 1, which reflects these results. The average accuracy levels out for both sequence lengths at a low embedding dimension but is higher for length 6 sequences.

Notably, accuracy saturates at relatively low embedding dimensions, which suggests that task success requires only limited representational capacity. To understand what additional capacity provides, we next examine internal representations.

Table 1: Average accuracy of the agents by length and number of iterations.

|          | Length 6   | Length 8   |
|----------|------------|------------|
| 1M iters | **96.7%**  | N/A        |
| 2M iters | N/A        | 74.3%      |
| 10M iters| 95.5%      | **90.7%**  |

### 5.2 REPRESENTATION OF GLOBAL ORDERING

Even though agents only needed to predict single swaps, most developed a coherent representation of the *entire* sequence ordering in the last row of the attention matrix. We quantified this with the proportion of non-inversions (see metric 2). For agents trained on sequences of length 6, the average proportion of non-inversions increases as the embedding dimension increases but levels out at 87% around embedding dimension of 30. This means there are on average only around 2 inversions between the input ordering and the attention output ordering.

For agents trained on sequences of length 8, the trend still exists, but it is not as strong because some agents never converge on a solution with 100% accuracy. If these agents are ignored, we see a similar result to the agents trained on length 6 sequences, where the average proportion of non-inversions increases until an embedding dimension of around 30 at which point it levels out at 78%.

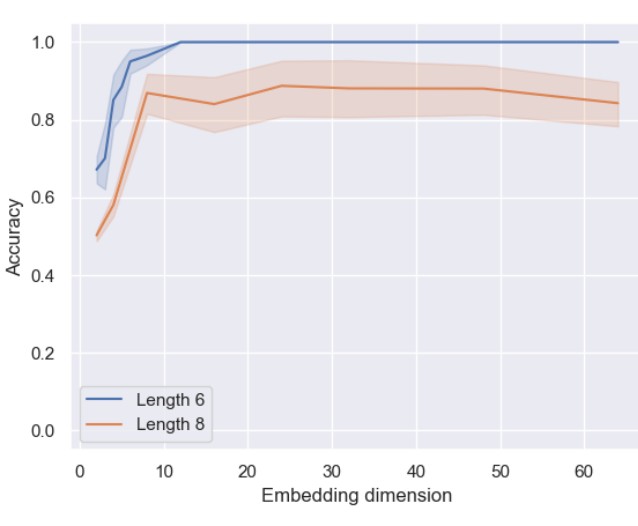

Figure 1: Accuracy vs. embedding dimension for length 6 and length 8 agents. Almost all agents achieve 100% accuracy for length 6, but this is not the case for length 8, where agents either achieve very high accuracy or get stuck at much lower accuracy. The mean and 95% confidence intervals are shown.

The agents with a lower accuracy only reach an average of 57%, close the 50% which would be expected for a random sequence. Figure 2 shows this trend, where the proportion of non-inversions is close to 50% at a low embedding dimension and increases as embedding dimension increases until it levels out, providing evidence for Observation 1 (see Appendix A.3 for additional visualizations of attention weights). Crucially, this metric increases well after the accuracy has already saturated. Larger embeddings therefore make its internal ordering representation stronger and more reliable.

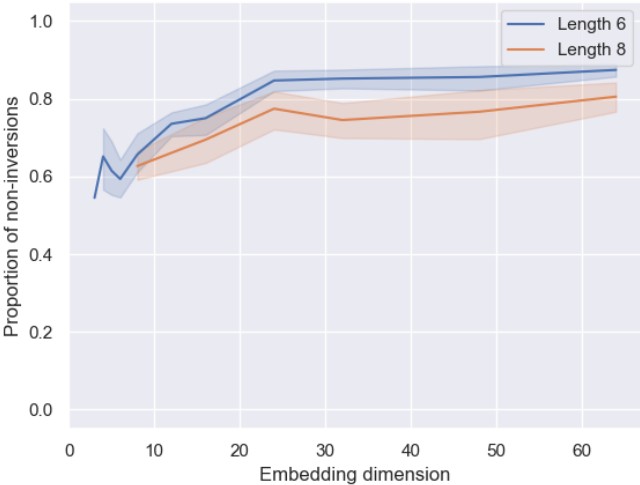

Figure 2: Proportion of non-inversions vs. embedding dimension for length 6 and length 8 sequences where the model has near-perfect accuracy. This reaches over 80% on average for both sequences when the embedding dimension is large enough. The mean and 95% confidence intervals are shown.

## 5.3 DECISION MECHANISM

We also found strong evidence for Observation 2: agents typically selected swaps by identifying the most positive or most negative difference between consecutive values in the last attention row. We analyzed how often the move an agent choose to make was in the top $i$ predictions according to the largest gaps in the attention matrix (metric 3). Examples of attention weight visualizations for high- and low-performing agents are shown in Appendix A.3. Across high-accuracy agents, 76–77% of moves matched the top-1 prediction, and more than 90% matched the top-2 (Table 2, Figure 3).

As with ordering fidelity, swap prediction alignment seemed to increase with embedding dimension until leveling off at around dimension 30. This shows that higher embedding dimensions not only strengthen global representations but also sharpen the decision rule based on them.

## 5.4 FAILURE MODES

Beyond aggregate accuracy metrics, we examined qualitative failure modes of low-dimension agents (Appendix A.3 includes representative training loss curves). A common pattern was what we term the "local greedy trap": swapping the most obviously incorrect local pair due to a failure to recognize that fixing a larger global inversion sometimes required temporarily increasing the number of local inversions. This trap is especially telling because it indicates that the model had not formed a coherent global ordering representation; instead, it relied on a heuristic driven purely by local differences in value embeddings. The appearance of this failure mode decreases with higher embedding dimensions, suggesting that greater representational capacity allows simultaneous encoding of both global and local order features.

Table 2: The proportion of moves an agent with high accuracy chooses which are the first/second largest gap in the last row of the attention matrix.

|  | Top 1 | Top 2 |
| --- | --- | --- |
| Length 6 | 76.2% | **94.0%** |
| Length 8 | **76.8%** | 92.5% |

This table includes almost all agents trained on length 6 sequences because they are almost all fully accurate. However, many length 8 agents never reach full accuracy so they are left out. The low-accuracy length 8 agents only have a top-2 proportion of 54.1%, which is much lower than the accurate agents. Figure 3 shows the proportion of correct top 1 and top 2 predictions as embedding dimension changes for the high accuracy agents. Similar to the proportion of non-inversions, these proportions increase with the embedding dimension until they level out at an embedding dimension of around 30.

## 5.5 CROSS-METRIC CONSISTENCY

Our two observations for how the model should act are very related. If the global ordering observation is true, then choosing the move with the most negative difference will always lead to a correct swap. Because of this, it makes sense that the performance of these two mechanisms against the embedding dimension are strongly related. This relationship can be seen in Figure 4, where there is a fairly strong association between the metrics used to evaluate Observation 1 and Observation 2. This suggests that either metric could serve as a proxy for the other in similar studies.

## 6 CONCLUSION

We studied how embedding dimension affects the emergence of interpretable internal mechanisms in small transformers trained with PPO to perform bubble-sort–style adjacent swaps. Across hundreds of agents, we found two consistent patterns: the final row of the attention weights encodes a global ordering of tokens, and the chosen swap corresponds to the largest adjacent difference in that ordering. While accuracy saturates at relatively low embedding dimensions, the faithfulness

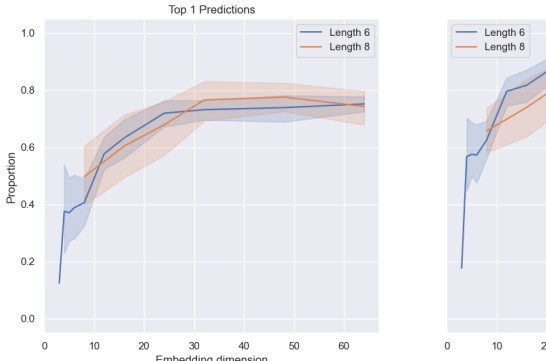

Figure 3: Proportion of moves which are the top 1/2 prediction according to our observation vs. embedding dimension for length 6 and length 8 sequences where the model has near-perfect accuracy. Almost all moves are chosen according to this observation, with the top 2 proportion reaching above 90% for a large enough embedding dimension. The mean and 95% confidence intervals are shown.

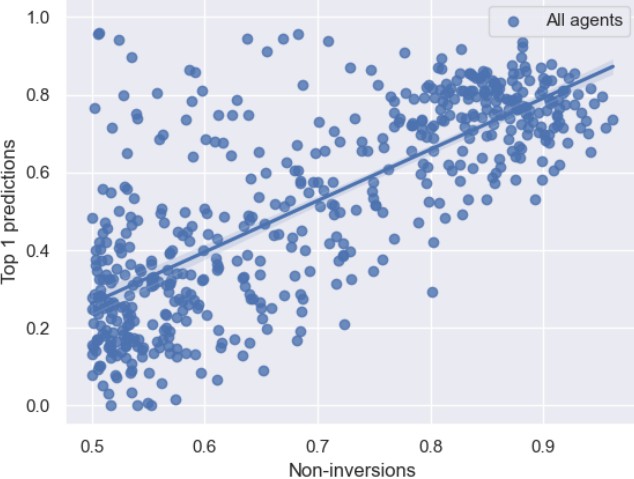

Figure 4: The proportion of top 1 predictions vs. proportion of non-inversions for all agents. These two metrics have an $r^2$ value of 0.56, so there is a fairly strong correlation between the two values. The line of best fit and 95% confidence interval for the parameters of this line are shown.

and robustness of these mechanisms continue to improve up to dimension around 30. Thus, capacity improves representation quality, providing quantitative evidence that transformers build simple, structured world models even in constrained RL settings.

## 6.1 DISCUSSION

The observation that representation quality continues to improve well after accuracy saturates has several implications. First, it highlights the importance of probing internal representations directly rather than relying solely on performance metrics, especially when evaluating model capacity. In safety-critical or interpretability-focused contexts, a more structured internal representation may be preferable even if accuracy is unchanged, as it can make the model's decision process more predictable and transparent.

From a scaling perspective, our findings parallel the idea of "capability plateaus" in large-scale models: increases in size yield diminishing returns on benchmark scores but continue to shape the internal geometry of the learned representation space. This reshaping may have downstream benefits, such as improved transfer to longer sequences, greater robustness to noise, or better alignment with human-interpretable concepts. In the RL setting, it could also reduce policy brittleness, as more structured latent states may generalize more smoothly to out-of-distribution configurations.

The main observation from our results is that the accuracy of the models levels out at very close to 100% at a very low embedding dimension (around 6 for length 6 sequences trained for 10M timesteps), but the metrics discussed above increase well beyond that (around 30 for the same agents). This suggests that in our experiments, the bubble sort circuit is, in some sense, the easiest and most effective solution for the agent to converge to. The fact that higher capacity leads to more consistent circuit formation across seeds also hints at a practical trade-off: overprovisioning model width might be a cheap way to make learned policies more interpretable without any loss in performance.

Furthermore, our observation that the last row of the attention matrix encodes the global ordering is not just a high-level correlation. It is a direct consequence of the model learning a structured embedding space. For the attention weights to produce a monotonic sequence, the underlying key vectors must themselves be arranged in an orderly fashion. Since these vectors are derived from the initial token embeddings, this implies that the embedding matrix has learned a representation where tokens with a higher numerical value are situated in a consistent direction in the embedding space.

## 6.2 APPLICABILITY

Our analysis demonstrates that even very small transformers, when trained in RL settings, naturally converge to simple, interpretable circuits for solving sorting tasks. This supports the idea that transformers, under constrained conditions, expose internal mechanisms that are easy to study. Similar approaches could be applied to other algorithmic tasks (e.g., graph problems, dynamic programming), where the attention matrix may again reveal hidden internal structure.

## 6.3 LIMITATIONS

This study is limited both in scope and scale. We focused only on adjacent transposition sorting with 1–2 layer, single-head transformers. More complex architectures—such as deeper models, multi-head attention, or richer environments—were not explored. This narrow scope, while useful for isolating core mechanisms, restricts the generalizability of our results to more complex, multi-layered architectures or real-world applications. Computational budget also restricted our ability to test broader ranges of sequence lengths or larger training runs. As such, our findings should be viewed as evidence of a general trend, not as an exhaustive characterization.

## 6.4 FURTHER WORK

Several extensions follow naturally. One direction is to relax the adjacency constraint and allow arbitrary in-place swaps; this could reveal whether transformers discover known efficient algorithms (e.g., merge sort) or converge to novel heuristics.

Additionally, we could introduce stochasticity into the environment, such as noisy token values or partial observability, to test whether the same ordering circuit emerges under uncertainty. Another direction is to explore hybrid training setups, where a model is pre-trained in a supervised manner on optimal swaps before being fine-tuned with RL; this could reveal whether the ordering representation is more robust when acquired via imitation or exploration.

Further, introducing multi-head attention would allow us to examine whether different heads naturally specialize in complementary subproblems (e.g., local comparison vs. global structure). It would also be valuable to test the robustness of these circuits under pruning or quantization, which could preserve interpretability while reducing model size. Lastly, applying similar analyses to structurally richer algorithmic tasks, such as graph algorithms, constraint solvers, or an algebra problem, could test the generality of the capacity–representation link and might uncover new, reusable circuit motifs.

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

## A  APPENDIX

### A.1  MODEL

The transformer architecture was first introduced in the paper *Attention Is All You Need* and was able to achieve state of the art results for translation tasks Vaswani et al. (2017). Since then, transformers have been applied to a wide range of machine learning tasks across language, vision, and other domains. The core mechanism in the architecture is the attention function, which calculates output vectors from input queries, keys, and values. In self-attention, the queries, keys, and values are all calculated from the same input sequence.

The model used in this experiment is a stripped down transformer with an embedding layer, a single-head self-attention block, and a linear layer. The toy problem we considered was extremely simple and could conceivably be solved with 100% accuracy by almost any sufficiently complex model. Our goal was to isolate attention weights to gain insight about how the problem was solved, so we took away as much of the rest of the model as possible. The main parameter of the model which was controlled to see how the results changed was the embedding dimension, which was also equal to the dimension of the keys/queries.

### A.2  TRAINING ALGORITHM

PPO is a RL algorithm which is simpler and more performant than many other leading algorithms. It does this by optimizing a lower bound of the policy performance created from clipped probability ratios. This can be implemented with an Actor-Critic framework, where the actor chooses actions

and the critic estimates the value function. The algorithm alternates between running the policy and optimizing for multiple epochs based on the results Schulman et al. (2017).

Our model is given a fixed-length permutation as a series of tokens (with no predefined numerical value) and output a single index representing a transposition which should make the input permutation closer to being sorted. This is trained using the PPO RL algorithm Schulman et al. (2017) and a Gym environment Brockman et al. (2016). The agent receives a reward of $1$ if the permutation is sorted after making a transposition and a reward of $-0.001$ otherwise. The initial permutation is randomized and each run terminates when either the permutation is fully sorted or 1000 transpositions have been made. The clipped value loss of the PPO algorithm Schulman et al. (2017) during an example training run can be seen in Figure 8.

Table 3: PPO parameters used while training the agents.

| Parameter | Value |
|---|---|
| Max steps for Gym environment | 1000 |
| Total timesteps | 1M/2M/10M |
| Learning rate | 2.5e-4 |
| Number of environments | 8 |
| Number of steps per policy rollout | 128 |
| Discount factor | 0.99 |
| General advantage estimation lambda | 0.95 |
| Number of minibatches | 4 |
| Surrogate clipping coefficient | 0.1 |
| Entropy coefficient | 0.01 |
| Value function coefficient | 0.5 |
| Maximum norm for gradient clipping | 0.5 |

## A.3 ADDITIONAL FIGURES

One of our main observations from this paper is that the final row of the attention weight matrix represents the global ordering of the input sequence. Figure 5 shows an agent's attention weight matrix when the input is fully ordered. If Observation 1 held, the last row of this matrix would contain fully ordered elements, which is exactly what happens.

This same phenomenon can also be visualized by looking at the spread of attention weight values corresponding to each token in the input sequence. Figures 6 and 7 show this for two example agents. Each figure shows a violin plot of the attention weight values of every token. For example, the violin above 2 shows the spread of last-row attention weights of the agent corresponding to the column where the input token was 2. A wider violin corresponds to a higher density of values. Both Figure 6 and Figure 7 show monotonic averages of the attention values, which is exactly what we would expect to see if the agent learned the global ordering of the tokens. It is important to note again that the model is never given the numerical values of the tokens, but is able to figure out their relative ordering just from the training process (with a large enough embedding dimension). Figure 7 has a much wider spread of values for each token, and this increased variation explains why it gets the ordering wrong more often.

The clipped value loss of an example training of an agent is shown in Figure 8. This is the loss for an agent which was able to achieve 100% accuracy, so the loss converges to 0. For agents which never converge to a fully accurate solution, the loss would likely not stabilize. This happens more when the sequence length is 8 because the search space to find correct solutions is so much larger.

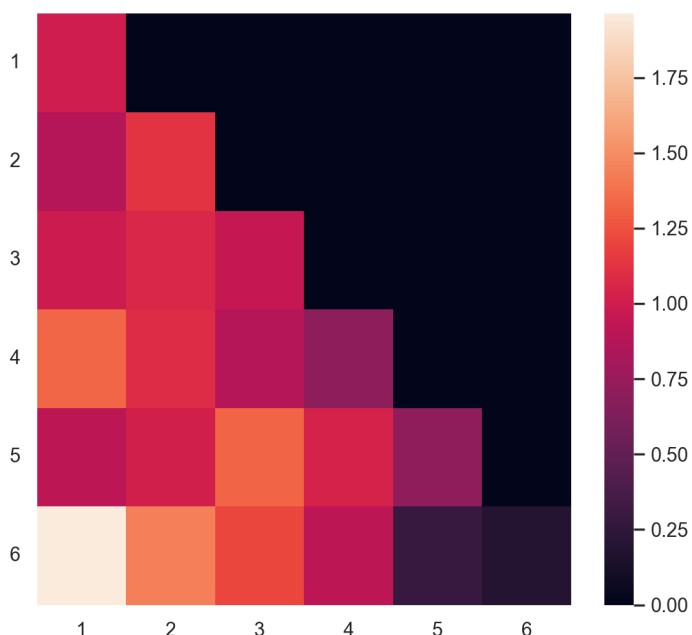

Figure 5: Visualization of the attention weights for a fully ordered permutation (using the same agent as in Figure 6). The weights in the last row are ordered just like the permutation, showing that for this agent and permutation there are no inversions.

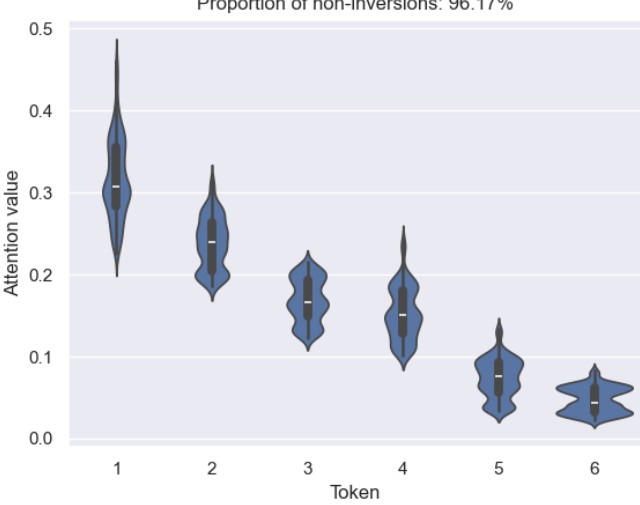

Figure 6: Attention weights by token for the agent with fewest average inversions.

702
703
704
705
706
707
708
709
710
711
712
713
714
715
716
717
718
719
720
721
722
723
724
725

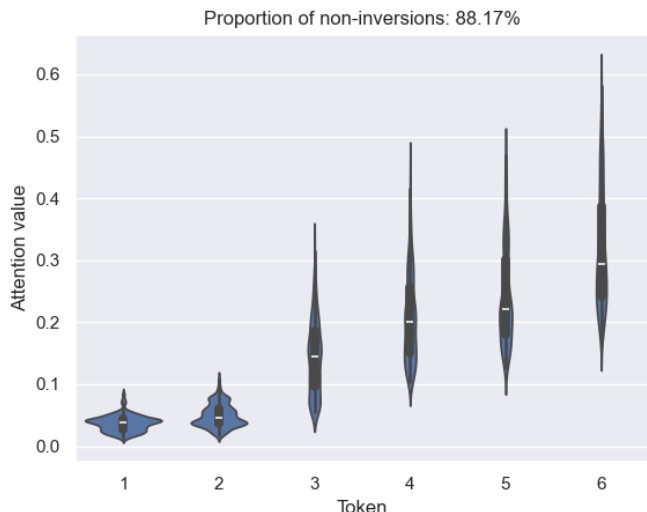

Figure 7: Attention weights by token for a random agent with high accuracy.

726
727
728
729
730
731
732
733
734
735
736
737
738
739
740
741
742
743
744
745
746
747
748
749
750
751
752
753
754
755

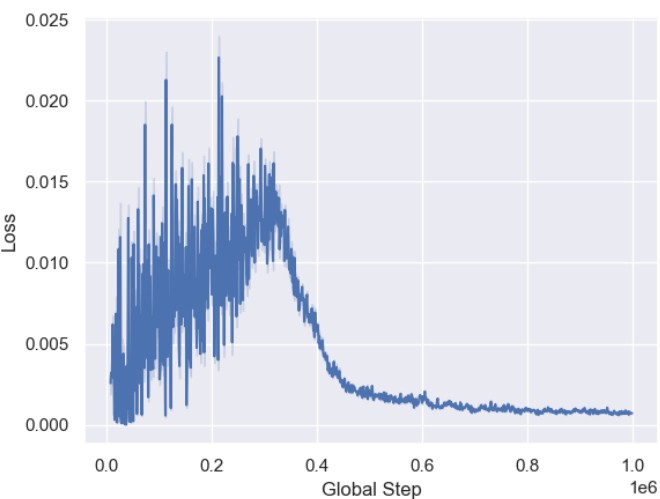

Figure 8: Clipped value loss vs. global step in an example training of an agent on a length 6 sequence.

