# OpenReview forum: "Higher Embedding Dimension Creates a Stronger World Model for a Simple Sorting Task"
_ICLR.cc/2026/Conference — Submitted to ICLR 2026_

### Official Review · Reviewer_GxPv · 2025-10-29

**Soundness:** 1
**Presentation:** 1
**Contribution:** 1
**Rating:** 0
**Confidence:** 5

**Summary:**

The paper aims to study empirical properties of Transformers that emerge in a sorting task setting. It include an empirical setup involving small models and short sequences (of lengths 6 and 8). The models were optimized with an RL objective with the goal of sorting the given input sequences.

The results presented in the paper suggest that single attention head, single layer Transformer models with embedding dimension less than 16 struggle to sort sequences of length 6 and 8. In addition, sorting sequences of length 8 was observed to be more challenging and require a higher embedding dimension.

**Strengths:**

Generally, the idea of finding a simple yet interesting task to study some aspect of a method we don't fully understand can lead to interesting discoveries.
Studying sorting algorithms could possibly lead to interesting observations in the context of Transformer mechanistic interpretability.

**Weaknesses:**

`W1`: The paper lacks a coherent structure. The preliminaries section includes fractured pieces of information with no direct flow, while discussing motivation as well.
The methods section opens with a description of an empirical setup and implementation details, which should naturally be described in the "experiments" section. The methods section should only include a precise description of the proposed method, focusing on the novel specifics.


`W2`: The methods section does not describe any novel method, but rather an experimental setup.


`W3`: The related works section fails to appropriately situate the paper in existing literature.
Specifically, while the discussion refers to some prior works, it does not discuss how these works are related to the paper, what is the gap addressed by this paper that was missing in prior works, etc.


`W4`: The literature review is insufficient. For example:
> The question of how model capacity—especially embedding size—affects representation quality has been widely studied (line 93)

The included discussion does not include sufficient relevant literature, which is expected given the "widely studied" statement.


`W5`: The claim
> We introduce sorting as a reinforcement learning testbed for mechanistic interpretability

is not sufficiently justified. While the paper proposes sorting as a testbed, the evidence are insufficient.


`W6`: the empirical setup involves extremely short sequences (6-8) and tiny model sizes (up to dim 128), which are rarely used in any practical setup. Thus, it is unclear to what extent the result are meaningful or representative of any practical model behavior.

Furthermore, to establish observations 1 and 2, an extensive study should be conducted, involving longer sequences, larger models, and studying the conditions that facilitate / enable these observed behaviors.
Otherwise, the observation should limit the scope to sequences of length 6-8, and the appropriate model sizes. In such a case, it is unclear what is the contribution of such an observation.


`W7`: Figure sizes are inconsistent, label sizes are too small, and figure sizes are generally disproportionally large.


`W8`:
> The observation that representation quality continues to improve well after accuracy saturates has several implications. (line 427)

The evidence in the paper is insufficient for supporting such a claim. Again, to establish such a general claim, outside the limited scope of extremely short sequences and small models, an extensive empirical study must be conducted.


`W9`: The paper lacks formal, precise formulations of the objects at the center of discussion. For example, in lines 220-230, it would be clearer to formally establish a notation and present the statements in a formal, precise mathematical language.
This extends beyond this single example, and is a general theme throughout the paper.

`W10`:
> From a scaling perspective, our findings parallel the idea of “capability plateaus” in large-scale models: increases in size yield diminishing returns on benchmark scores but continue to shape the internal geometry of the learned representation space (line 432)

The discussion on scaling is not supported by the evidence presented in the paper.


`W11`: The paper lack accuracy and precision throughout. As an example:
> This suggests that in our experiments, the bubble sort circuit is, in some sense, the easiest and most effective solution for the agent to converge to (line 441)

This conclusion is not supported by the evidence in the paper. The claim is phrased in a careless manner.


`W12`: There are indications for a significant use of LLMs, yet LLM usage was not disclosed by the authors, as required by the ICLR 2026 guidelines.

**Questions:**

`Q1`: Why did you choose to optimize the model with RL? In a sorting task setup the true labels are available. What is the role of RL here? Why is it interesting?

`Q2`: What is the structure of the sequences in the dataset (to be sorted)? integers? real numbers? How do you chose / sample them?

---

### Official Review · Reviewer_PrWz · 2025-10-31

**Soundness:** 2
**Presentation:** 3
**Contribution:** 3
**Rating:** 2
**Confidence:** 3

**Summary:**

This paper investigates how embedding dimension affects internal "world model" emergence in RL-trained transformers for bubble-sort-style swaps. Models achieve high accuracy even with small embedding dimensions, but larger dimensions produce more faithful, consistent, and robust internal representations, enhancing structured encoding and interpretability. Hundreds of experiments reveal two mechanisms: the last row of the attention weight matrix monotonically encodes token global order, and selected transpositions align with the largest adjacent differences of these encoded values. Results provide quantitative evidence that transformers build structured world models, with larger embedding dimensions improving representation quality beyond end performance. Metrics and analyses are released for similar algorithmic tasks.

**Strengths:**

1. Enhanced Representation Quality: Larger embedding dimensions strengthen structured internal representations—boosting global order encoding fidelity (non-inversion proportion reaches 87% for length-6 sequences) and sharpening swap decision rules (76–77% top-1 swap alignment). This goes beyond mere accuracy, enabling more robust and consistent world-model formation.​

2. Strong Interpretability: The study identifies two clear, consistent mechanisms (global order in attention weights, largest adjacent difference for swaps) that explain model behavior. This makes the transformer’s internal sorting logic interpretable, addressing the "black box" issue common in neural networks, especially for algorithmic tasks.​

3. Valuable Testbed & Metrics: It establishes sorting as a robust RL testbed for mechanistic interpretability and releases quantitative metrics (sorting accuracy, non-inversion proportion, swap-rule alignment). These resources enable reproducible research on how model design impacts internal representations for similar algorithmic tasks.

**Weaknesses:**

1. Task Specificity: The study focuses solely on a simple bubble-sort-style adjacent swap task with small sequence lengths (6–8). Results may not generalize to complex algorithmic tasks (e.g., merge sort, graph algorithms) or real-world sequence tasks (e.g., text processing), limiting broader applicability.​

2. Embedding Dimension Saturation: Beyond ~30 embedding dimensions, improvements in representation quality (non-inversion proportion, swap alignment) level off. This means excessive embedding dimension increases waste computational resources without meaningful gains, lacking guidance on optimal dimension for diverse task scales.​

3. Sequence Length Limitations: Length-8 sequences see far lower full-accuracy rates (37.4% vs. 99.2% for length-6) even with high embedding dimensions. The model struggles with larger combinatorial state spaces, revealing poor scalability to longer sequences critical for real-world use cases.​

4. Over-Reliance on Minimal Architecture: Experiments use stripped-down transformers (single-head attention, no MLPs) to isolate embedding dimension effects. Findings may not apply to modern multi-head, deep transformers—failing to address how embedding dimension interacts with complex architectural components in practical models.​
5. Limited Failure Mode Analysis: While identifying the "local greedy trap" for low-dimension models, the study lacks in-depth exploration of other failure modes (e.g., sensitivity to initial permutations, training instability in high-dimension models). This incomplete failure analysis hinders troubleshooting when adapting the findings to new tasks.​

**Questions:**

1. Task Specificity: The study focuses solely on a simple bubble-sort-style adjacent swap task with small sequence lengths (6–8). Results may not generalize to complex algorithmic tasks (e.g., merge sort, graph algorithms) or real-world sequence tasks (e.g., text processing), limiting broader applicability.​

2. Embedding Dimension Saturation: Beyond ~30 embedding dimensions, improvements in representation quality (non-inversion proportion, swap alignment) level off. This means excessive embedding dimension increases waste computational resources without meaningful gains, lacking guidance on optimal dimension for diverse task scales.​

3. Sequence Length Limitations: Length-8 sequences see far lower full-accuracy rates (37.4% vs. 99.2% for length-6) even with high embedding dimensions. The model struggles with larger combinatorial state spaces, revealing poor scalability to longer sequences critical for real-world use cases.​

4. Over-Reliance on Minimal Architecture: Experiments use stripped-down transformers (single-head attention, no MLPs) to isolate embedding dimension effects. Findings may not apply to modern multi-head, deep transformers—failing to address how embedding dimension interacts with complex architectural components in practical models.​
5. Limited Failure Mode Analysis: While identifying the "local greedy trap" for low-dimension models, the study lacks in-depth exploration of other failure modes (e.g., sensitivity to initial permutations, training instability in high-dimension models). This incomplete failure analysis hinders troubleshooting when adapting the findings to new tasks.​

---

### Official Review · Reviewer_zjjy · 2025-10-31

**Soundness:** 3
**Presentation:** 2
**Contribution:** 1
**Rating:** 2
**Confidence:** 4

**Summary:**

This paper studies LLMs trained using reinforcement learning to run steps of the bubble sort algorithm. They find a higher embedding dimension improves performance and leads to the state of the system being stored in attention weights. They also identify patterns across the experiments which suggest a simple bubble sort algorithm:

- For a given input permutation, the LLM produces internal representations which are monotonic in the sorted order. (I.e., internal representations for a particular input token is small when the input token is smaller).
- The pair of tokens the LLM chooses to swap are the two with the biggest gap between them (i.e., a very large number before a very small number). They call this a “difference-based swap rule”.

**Strengths:**

- The exposition was clear.
- I find it somewhat interesting that the LLMs did indeed seem to record all relevant information about the state in internal representations.

**Weaknesses:**

- There were some omissions in the way the learning task was described.
    - Where the input tokens numeric or text? If they were already numeric, step one of the discovered algorithm could just be the identity function, or any monotonic transform?
- The paper does not offer new evidence as to why increasing the embedding dimension leads to increased fidelity of state space representation, beyond observing that this is true. I would expect more discussion or exploration of why this might be the case. If the answer is that models with more parameters have more expressive power, the result does not seem particularly surprising or interesting.
- Moreover, the paper does little to suggest how the insights from the bubble sort algorithm will generalize to other contexts, or how to think of their task as an instance of a large class of interesting state-tracking problems.

**Questions:**

- Why do you think the transformer learns the difference-based swap rule? It is particularly efficient?
- I didn’t understand Section 5.4. In particular, why is it true that “ swapping the most obviously incorrect local pair” is a bad idea? Any correct swap should make progress towards the sorted end state, no?

---

### Official Review · Reviewer_aDUg · 2025-11-04

**Soundness:** 2
**Presentation:** 1
**Contribution:** 2
**Rating:** 2
**Confidence:** 4

**Summary:**

In this work, the authors investigate the internal mechanisms by which transformers learn to perform bubble sorting when trained via reinforcement learning. Their results show that, with a sufficiently large embedding dimension, transformers can internally develop interpretable structural representations. Specifically, the last row of the attention weights encodes the global ordering of the tokens, while the learned value matrix, together with the output head, is used to decode the transposition corresponding to the pair of adjacent attention values with the largest difference.

**Strengths:**

The paper’s empirical findings are both interesting and somewhat surprising. They show that models trained via gradient descent can naturally converge to interpretable solutions when the model’s capacity is sufficiently large. While this observation is broadly consistent with previous work (e.g., [1]), it is demonstrated here in a new and perhaps more advanced setting.

[1] Yang et al., Emergent Symbolic Mechanisms Support Abstract Reasoning in Large Language Models

**Weaknesses:**

- The wording of this paper could be improved. The terms “world model” and “agent” are not clearly defined, which makes their use vague in a technical context. Typically, these terms are used in more advanced AI systems, such as robotics or large language models, and using them to refer to a transformer trained under reinforcement learning on synthetic tasks may be misleading.

- Additionally, the experimental details are not sufficiently clear. It is difficult to understand the precise inputs and outputs of the transformer. For example, is the transformer performing a sequence-to-sequence mapping from an unsorted sequence to a target sequence of transpositions, or does it autoregressively generate the transposition sequence conditioned on the unsorted sequence? Similarly, the design of the reinforcement learning environment is unclear, including how the model interacts with the environment. I recommend that the authors use clear notations or formulas to illustrate the model architecture, the inputs and outputs, the RL environment (e.g., using MDP language), and the evaluation metrics.

- Despite the interesting empirical observations in a controlled setting, it remains unclear whether these findings generalize to more practical scenarios, such as multi-layer attention, or to more challenging tasks. Given the largely empirical nature of the work, its significance and contribution to the broader field appear limited.

**Questions:**

1. Does teacher forcing also lead to the emergence of similar interpretable patterns?

2. In addition to embedding dimension, are there other hyperparameters that are critical for the formation of interpretable representations? Could the authors provide insights into whether this phenomenon is architecture-dependent (depth, width, head number or even transformers vs RNNs)?

3. Were all the tested models trained from scratch using RL, or were there any pretraining?

---

### Meta-Review · Area_Chair_usow · 2026-01-08

**Summary:**

The submission investigates the emergence of internal representations in transformers trained via reinforcement learning to perform a bubble-sort task on short sequences. The authors claim that while low embedding dimensions are sufficient for accuracy, higher dimensions foster interpretable world models, specifically encoding global order in attention weights. The paper positions this setup as a testbed for mechanistic interpretability. However, the study is strictly empirical and limited to a minimalist architecture.

**Reviewer Concerns:**

Outstanding Concerns:
1. Experimental Scope and Scalability: All reviewers criticized the extremely limited scope. The study relies on sequence lengths of 6-8 and a stripped-down architecture (single-head, no MLP). Reviewers argued that these toy constraints make it impossible to generalize findings to practical models or complex algorithmic tasks.

2. Methodological Validity and Justification: Reviewer GxPv fundamentally questioned the use of RL for a sorting task where ground truth is available, noting the lack of comparison to supervised baselines. Reviewer zjjy noted that the paper observes that capacity helps but fails to provide theoretical insight into why.

**Reviewer Scores:**

no changes due to no response

---

### Decision · Program_Chairs · 2026-01-26

Reject